Neuroprotective effects of essential oils in animal models of Alzheimer’s and Parkinson’s disease: a systematic review

Macedo Adrielle do Espírito Santos 1
Ferreira Thayná Moraes 1
Krejcová Lane Viana 1
Rocha Fernando Allan de Farias 1
da Silva Joyce Kelly R. 2
Pedrosa Laís Resque Russo 1
http://orcid.org/0000-0002-3646-7847 Gomes Bruno Duarte 1 3 brunodgomes@ufpa.br
1 Laboratório de Neurofisiologia Eduardo Oswaldo Cruz, Instituto de Ciências Biológicas, Universidade Federal do Pará , Belém, Pará , Brazil
2 Laboratório de Bioprospecção de Inovação Tecnológica de Produtos Naturais da Amazônia, Instituto de Ciências Biológicas, Universidade Federal do Pará , Belém, Pará , Brazil
3 Laboratório de Simulação e Biologia Computacional, Centro de Computação de Alto Desempenho e Inteligência Artificial, Universidade Federal do Pará , Belém , Brazil
Parker Matthew
Electronic publication date: 2025 Jul 10
Publication date: 2025
Volume: 13
Electronic Location ID: e19643
Received 2024 Nov 13; Accepted 2025 Jun 2
Copyright: © 2025 Macedo et al.
Copyright year: 2025
Copyright holder: Macedo et al.
License: This is an open access article distributed under the terms of the Creative Commons Attribution License, which permits unrestricted use, distribution, reproduction and adaptation in any medium and for any purpose provided that it is properly attributed. For attribution, the original author(s), title, publication source (PeerJ) and either DOI or URL of the article must be cited.
License URL: https://creativecommons.org/licenses/by/4.0/

Keywords: Essential oils, Neuroprotection, Alzheimer’s disease, Parkinson’s disease, Systematic review

Funding: The authors received no funding for this work.

==============================
Essential oils (EOs), derived from aromatic plants, have garnered significant attention for their potential neuroprotective properties in neurodegenerative diseases. This systematic review evaluates recent advancements in understanding the neuroprotective role of EOs against Alzheimer’s disease (AD) and Parkinson’s disease (PD). Following PRISMA guidelines, we conducted a comprehensive literature search across three major databases (PubMed, Virtual Health Library, and Web of Science) from inception to January 2024, resulting in thirteen high-quality in vivo studies for qualitative analysis. The review assessed various EOs, with hydrodistillation being the predominant extraction method (66.66% of studies). Studies primarily utilized Wistar rats (46.15%) and various mouse strains, employing diverse disease induction methods including β-amyloid administration (30.7% of AD models), rotenone (7.7% of PD models), and 6-hydroxydopamine (7.7% of PD models). Administration routes varied, with oral administration being most common (38.4%), with gavage and inhalation each accounting for 23.1% of studies. Key findings revealed that EOs exhibit multifaceted neuroprotective mechanisms. In AD models (69.3% of studies), EOs reduced oxidative stress markers, decreased pro-inflammatory cytokine levels, and increased neuroprotective protein expression. In PD models (30.7% of studies), EOs demonstrated significant dopaminergic neuroprotection, with improvements in behavioral outcomes. Behavioral assessments showed consistent enhancements in memory, learning, and motor functions across studies. The systematic analysis provides compelling evidence for EOs’ neuroprotective efficacy, particularly in early-stage intervention. However, limitations include the predominance of animal studies, variability in dosing, and administration methods. The most promising EOs identified were from Pinus halepensis, Citrus limon, and Acorus species, showing particular efficacy in reducing cognitive deficits and oxidative stress. Chemical analysis revealed that compounds such as α-pinene, limonene, and β-caryophyllene were predominantly responsible for the observed therapeutic effects. The molecular mechanisms underlying these effects included modulation of cholinergic transmission, reduction of amyloid-β aggregation, and enhancement of antioxidant enzyme activities. These findings suggest that EOs could serve as valuable complementary therapeutic agents, particularly when standardized for specific bioactive compounds. Future research should focus on standardizing EO compositions, conducting human clinical trials to establish safety and efficacy profiles, and investigating potential synergistic effects with conventional treatments.

Introduction

Neurodegenerative diseases represent one of the greatest challenges in modern medicine, with Alzheimer’s disease (AD) and Parkinson’s disease (PD) at the forefront of this global health crisis. According to a WHO and Lancet Neurology study, over 3 billion people worldwide have neurological conditions (Steinmetz et al., 2024). For AD specifically, an estimated 6.9 million older adults are living with AD in the United States alone, with another 200,000 people under age 65 having younger-onset AD. For Parkinson’s disease, nearly 90,000 people are diagnosed every year in the U.S., and 1.2 million people in the U.S. will be living with Parkinson’s by 2030. The economic burden of these conditions is staggering. In the United States alone, Americans spent $196 billion in direct medical costs for AD and related dementias in 2020, with an additional $254 billion in caregiver time (Nandi et al., 2024). For PD, the economic burden increases threefold as the disease progresses, primarily driven by hospitalizations, prescription medications, and indirect costs (Chaudhuri et al., 2024). These devastating conditions are characterized by progressive neuronal loss in specific brain regions, leading to profound cognitive and motor deficits that dramatically impact patients’ quality of life (Kwok, 2010). Despite significant advances in understanding their molecular mechanisms, current therapeutic approaches remain largely palliative, highlighting an urgent need for innovative treatment strategies.

Essential oils (EOs) have emerged as promising candidates in this therapeutic quest. These complex mixtures of volatile compounds, extracted from medicinal and aromatic plants, contain diverse bioactive components including terpenes, sesquiterpenes, and phenols (Koyama & Heinbockel, 2020). Their unique molecular properties, particularly their ability to cross the blood-brain barrier due to their lipophilic nature and low molecular weight (below 500 Da), position them as potential therapeutic agents (Fernandes et al., 2021). Their demonstrated antioxidant, anti-inflammatory, and neuroprotective properties in experimental models (Lee, Chu & Chiang, 2021; Banji, Banji & Srinivas, 2021) suggest potential therapeutic applications through multiple mechanisms, including reactive oxygen species neutralization, inflammatory pathway modulation, and neuronal plasticity enhancement (Liu et al., 2020).

The therapeutic potential of plant-derived compounds represents a paradigm shift in approaching neurodegenerative disease treatment. Furthermore, recent studies have revealed that EOs can modulate key pathological processes through multiple mechanisms, including neurotrophic factor expression, oxidative stress reduction, and neuroinflammation suppression (Postu et al., 2022; Gao et al., 2019). This multi-target approach aligns well with the complex pathophysiology of neurodegenerative disorders.

Given the promising preclinical evidence surrounding essential oils, a systematic analysis of the existing literature is crucial to better understand their therapeutic potential in neurodegenerative diseases. This review aims to critically examine and synthesize the available evidence on essential oils’ neuroprotective properties, helping to identify knowledge gaps and guide future research directions. By examining their biochemical properties and mechanisms of action, we aim to identify promising directions for developing EO-based interventions that could offer more effective and less invasive treatment options for neurodegenerative diseases. We hope to contribute to the growing body of knowledge that may ultimately lead to developing evidence-based complementary approaches for neurodegenerative conditions.

Materials and Methods

Protocol and registration

This systematic review was conducted under the Preferred Reporting Items for Systematic Reviews and Meta-Analyses (PRISMA) guidelines and was registered in a designated database (10.17605/OSF.IO/6MF78).

Information sources and search

Search descriptors were selected based on the Population/Problem, Interest, and Context (PICo) strategy. This review focused on experimental in vivo studies examining the neuroprotective role of essential oils (P) in neurodegenerative diseases (Co).

The databases searched included PubMed (US National Library of Medicine National Institutes of Health), Virtual Health Library, and Web of Science, utilizing the MeSH descriptors “essential oils,” “neuroprotection,” “Parkinson’s Disease,” “Alzheimer’s Disease,” “Huntington Disease,” and “Amyotrophic Lateral Sclerosis.” The search strategy was tailored to each database following the PICo inclusion criteria (detailed in the Supplemental Material). Additionally, search alerts were established to identify new studies.

Study selection and data collection

Studies published in English without date restrictions were considered. The inclusion criteria were experimental in vivo studies investigating the potential neuroprotective effects of essential oils on neurodegenerative diseases. Exclusions were made for reviews, case reports, clinical studies, and in vivo experimental studies not focusing on essential oils. Studies using plant extracts or derivatives instead of essential oils were also excluded. Below, are the eligibility criteria (inclusion/exclusion) in detail.

Inclusion criteria

Study design Experimental in vivo studies

Original research articles

Full-text articles available in English

Studies investigating neuroprotective effects of essential oils

Studies with clear methodological description

Experimental model Animal models (rodents)

Studies with appropriate control groups

Clear disease induction protocols for AD or PD

Intervention Essential oils as primary intervention

Clear description of EO source and extraction method

Specified dosage and administration route

Well-defined treatment duration

Outcomes Behavioral assessments

Biochemical analyses

Histopathological evaluations

Clear reporting of neuroprotective effects

Exclusion criteria

Publication type Reviews, meta-analyses, systematic reviews

Case reports or case series

Conference abstracts

Book chapters

Opinion articles

Study design In vitro studies

Ex vivo studies

Clinical studies

Studies using non-rodent models

Studies without control groups

Intervention Plant extracts or derivatives not classified as essential oils

Studies using essential oil components in isolation

Studies without clear description of essential oil composition

Combined interventions where EO effects cannot be isolated

For study selection, the Rayyan review assistance tool (https://rayyan.ai/) was employed to eliminate duplicates and filter out studies not meeting the eligibility criteria based on their titles or abstracts. The final selection was conducted by two independent reviewers (AM and TF), with any disagreements resolved by a third reviewer (LP).

Data extracted from the selected articles included publication year, origin, and compounds of the essential oils, methodologies, and primary outcomes related to their neuroprotective effects on the neurodegenerative diseases under study.

Risk of bias assessment

The risk of bias in individual studies was evaluated using the Systematic Review Center for Laboratory Animal Experimentation (SYRCLE) tool, adapted from the Cochrane Risk of Bias (RoB) tool for animal intervention studies (Hooijmans et al., 2014). This assessment was carried out by two independent reviewers (AM and TF) across ten domains to determine methodological quality: Sequence Generation, Baseline Characteristics, Allocation Concealment, Random Housing, Blinding (Performance Bias), Random Outcome Assessment, Blinding (Detection Bias), Incomplete Outcome Data, Selective Outcome Reporting, and Other Sources of Bias.

In the evaluation of methodological soundness, articles predominantly receiving “no” responses were deemed methodologically robust and indicative of a low risk of bias. Conversely, articles with a preponderance of “yes” responses were categorized as having a high risk of bias. Instances where the risk of bias could not be ascertained due to insufficient information were classified as having an “unclear risk.

To further elucidate the risk of bias, reviewers addressed additional questions: 1. Is there potential for the results to be biased?

2. Are there confounding factors that could obscure the interpretation of the results?

3. Is there a possibility that the study results are attributable to chance?

Results

Study selection

During the initial database screening, a total of seventy-seven articles were identified. Subsequent application of the Rayyan selection tool facilitated the exclusion of 30 duplicates and 23 articles that did not meet the eligibility criteria. This refinement process resulted in twenty-four articles being selected for full-text review. Of these, seven articles were excluded due to their reliance on in vivo experimental designs not aligned with our criteria, and one article was excluded because it utilized Drosophila as the animal model, which was outside the scope of our review. Additionally, three articles were excluded for not specifically addressing the neuroprotective effects of essential oils. Consequently, thirteen articles were deemed eligible for qualitative assessment, as illustrated in Fig. 1.

Figure 1 PRISMA flow diagram illustrating the systematic review selection process.

The diagram details the identification, screening, and inclusion phases of the literature search. From an initial pool of 77 articles identified through database searching, 30 duplicates were removed. Of the remaining 47 articles, 23 were excluded based on title and abstract screening. From the 24 full-text articles assessed for eligibility, 11 were excluded: seven due to incompatible in vivo experimental designs, one for using Drosophila as an animal model, and three for not specifically addressing essential oils’ neuroprotective effects. The final qualitative analysis included 13 articles that met all inclusion criteria and quality standards.

Study characteristics

The included studies exhibited variability in several key areas, including the type of essential oil (EO) investigated (Fig. 2). In terms of EO extraction methods, hydrodistillation was the most common, used in eight articles, accounting for 66.66% of the studies (Postu et al., 2019; Lee, Chu & Chiang, 2021; Gao et al., 2019; Beserra-Filho et al., 2019; Postu et al., 2022; de Campos et al., 2023; de Lucena et al., 2020; Beserra-Filho et al., 2022). Steam distillation was reported in one article, representing 7.7% (Banji, Banji & Srinivas, 2021), and supercritical carbon dioxide extraction was also noted in one article, comprising 7.7% (Xu et al., 2023). Three articles did not specify the extraction method used (Liu et al., 2020; Jayaraj et al., 2022; Aykac et al., 2022).

Figure 2 Methodological landscape of essential oil research in neurodegenerative disease models.

This diagram illustrates key experimental parameters across studies investigating essential oils’ neuroprotective effects in neurodegenerative disease models. It highlights four main domains: (1) Animal models—Wistar rats were used most frequently (46.15%), followed by Swiss mice (15.4%), transgenic mice (15.4%), and other strains such as Swiss albino, Kunming, and C57BL/6J; (2) Administration routes—oral administration was predominant (38.4%), followed by gavage (23.1%), inhalation (23.1%), intraperitoneal injection (7.7%), and intragastric administration (7.7%); (3) Essential oil extraction methods—hydrodistillation was most common (66.66%), with fewer studies using steam distillation (7.7%) or supercritical CO2 extraction (7.7%). Notably, 23.1% of studies did not specify their extraction method; and (4) Research focus—studies were divided between Alzheimer’s disease and Parkinson’s disease models.

The mode of EO administration varied across the studies: oral administration was documented in five articles (38.4%) (Banji, Banji & Srinivas, 2021; Jayaraj et al., 2022; Beserra-Filho et al., 2019; Aykac et al., 2022; Beserra-Filho et al., 2022), gavage in three articles (23.1%) (Lee, Chu & Chiang, 2021; Xu et al., 2023; de Lucena et al., 2020), inhalation in three articles (23.1%) (Postu et al., 2019; Liu et al., 2020; Postu et al., 2022), intraperitoneal administration in one article (7.7%) (de Campos et al., 2023), and intragastric administration in one article (7.7%) (Gao et al., 2019).

The research predominantly focused on AD and PD, representing 69.3% and 30.7% of the studies, respectively. AD induction methods included intracerebroventricular and intrahippocampal administration of β-amyloid (Postu et al., 2019; Lee, Chu & Chiang, 2021; Gao et al., 2019; Postu et al., 2022), intraperitoneal administration of aluminum chloride (Banji, Banji & Srinivas, 2021), intraperitoneal administration of scopolamine (Aykac et al., 2022; de Campos et al., 2023), and the use of transgenic mice as a disease model (Xu et al., 2023; Liu et al., 2020). PD models involved intraperitoneal administration of rotenone (Jayaraj et al., 2022), subcutaneous administration of reserpine (Beserra-Filho et al., 2022, 2019), and intracerebral administration of 6-hydroxydopamine (de Lucena et al., 2020).

Regarding animal models, the Wistar rat lineage was the most frequently used, present in six studies (46.15%) (Postu et al., 2019; Jayaraj et al., 2022; Postu et al., 2022; de Campos et al., 2023; Aykac et al., 2022). Swiss mice were utilized in two studies (15.4%) (Beserra-Filho et al., 2019, 2022), and transgenic mice were also used in two studies (15.4%) (Liu et al., 2020; Xu et al., 2023). Additionally, one study each employed Swiss albino, Kunming, and C57BL/6J lineages (Banji, Banji & Srinivas, 2021; Gao et al., 2019; Lee, Chu & Chiang, 2021).

Risk of bias within studies

The evaluation of methodological quality across ten domains revealed that most studies were classified as having a “low risk of bias,” indicating robust study methodologies among the included articles (Fig. 3). Notably, no studies were identified as possessing a high risk of bias, underscoring the methodological soundness of the research under review.

Figure 3 Summary of risk-of-bias assessment across all included studies presented as a stacked bar chart.

The chart provides an overview of judgments regarding risk-of-bias domains expressed as percentages across evaluated studies. The assessment covers ten key methodological domains: sequence generation, baseline characteristics, allocation concealment, random housing, blinding (performance bias), random outcome assessment, blinding (detection bias), incomplete outcome data, selective outcome reporting, and other sources of bias. Green bars represent low risk of bias, indicating robust methodological practices, while yellow bars signify unclear risk, suggesting areas where reporting could be improved. The absence of red bars indicates that no domains were classified as having a high risk of bias across the studies. This visualization demonstrates strong methodological quality in areas such as baseline characteristics and random housing, while highlighting domains like allocation concealment and blinding where transparency in reporting could be enhanced.

Specifically, the domains of “baseline characteristics,” “random housing,” “random outcome assessment,” “selective reporting,” and “other sources of bias” were deemed 100% adequate, reflecting optimal methodological practices (Fig. 4).

Figure 4 Risk-of-bias assessment for individual studies included in this systematic review presented across ten methodological domains following the SYRCLE tool criteria.

Each row represents an individual study, and columns represent different bias domains: sequence generation, baseline characteristics, allocation concealment, random housing, blinding (performance bias), random outcome assessment, blinding (detection bias), incomplete outcome data, selective outcome reporting, and other sources of bias. Green circles indicate low risk of bias, while yellow circles denote unclear risk. The visualization demonstrates that most studies maintained robust methodological standards, with a predominant low risk of bias across domains, though some areas show unclear risk assessment due to insufficient reporting of methodological details. This assessment highlights the overall quality of the included studies while identifying specific areas where reporting transparency could be improved. Studies: Aykac et al., 2022; Banji, Banji & Srinivas, 2021; Beserra-Filho et al., 2019, 2022; de Campos et al., 2023; de Lucena et al., 2020; Gao et al., 2019; Jayaraj et al., 2022; Lee, Chu & Chiang, 2021; Liu et al., 2020; Postu et al., 2019, 2022; Xu et al., 2023.

In the domain of random outcome assessment, 92.3% of the studies demonstrated methodological efficiency. However, one study presented ambiguity in methodological details, resulting in an “unclear risk” classification (Postu et al., 2019). Conversely, the domains of random sequence generation, allocation concealment, blinding of participants and personnel, and incomplete outcome data were categorized as having an “unclear risk of bias,” indicating areas where methodological clarity and rigor could be improved.

Results of individual studies and synthesis

In this systematic review, nine out of thirteen articles employed AD models to explore the neuroprotective role of essential oils (EOs) through their antioxidant effects, synaptic plasticity, and enhancement of learning and memory formation (Postu et al., 2019; Liu et al., 2020; Lee, Chu & Chiang, 2021; Gao et al., 2019; Xu et al., 2023; Postu et al., 2022; Lee, Chu & Chiang, 2021; de Campos et al., 2023; Aykac et al., 2022). Detailed results of these individual studies are presented in Table 1.

Table 1 Essential oils’ neuroprotective effects in AD and PD models.

Comprehensive overview of essential oils’ neuroprotective effects in experimental models of neurodegenerative diseases. The table summarizes key methodological aspects and outcomes from thirteen in vivo studies, including essential oil sources, chemical compositions, disease models, administration routes, behavioral and biochemical analyses, and main therapeutic findings. Studies are organized chronologically and categorized by disease type (Alzheimer’s disease and Parkinson’s disease).

Author	Essential oil and chemical composition	Disease model (Toxin/Dose/Duration/Administration Route)	Animal model (Species/Sex/Age/Weight)	Experimental groups	EO dosage	Behavioral tests	Experimental analyses	Oxidative stress markers	Inflammatory markers	Main results	
Aykac et al. (2022)	Origanum onites
Carvacrol (78.4%), g-Terpinene (6.9%) and p-Cymene (4.1%)	Alzheimer’s disease induced by Scopolamine (1 mg/kg, i.p., Daily until sacrifice)	Wistar albino rats, Females, Not specified, 200–250 g	-Control
-Scopolamine
-Scopolamine + galantamine
-EO
-Scopolamine + EO	Animals were treated with the essential oil orally. All treatments were administered daily at 9 a.m. until the end of the isolation period.	Locomotor activity test
Morris water maze test.
Novel object recognition test	Euthanasia and removal of brain tissue samples
Determination of AChE activity and measurement
of MDA and GSH levels in diferent brain tissues
Immunoblotting analysis
Protein–ligand docking	MDA (↓), GSH (↑)	COX-2 inhibition (↓), iNOS inhibition (↓), MPO inhibition (↓)	Significant relief of scopolamine-induced learning and memory impairments due to inhibition of acetylcholinesterase activity, attenuation of oxidative stress, and prevention of neuronal apoptosis in the hippocampus and frontal cortex.	
Banji, Banji & Srinivas (2021)	Curcuma longa L., Zingiberaceae
(Used Extract and essential oil (TE+EO))
Ar-turmerone (48,48%)	Alzheimer’s disease induced by Aluminum chloride (AlCl3) (40 mg/kg/day, i.p., 45 days)	Swiss albino mice, Males, 12 weeks, 30–35 g Swiss albino mice	-Control (vehicle 0.3% CMC)
-Positive control (treated with AlCl3; 40 mg/kg)
-TE+EO (25 mg/kg oral) + AlCl3 (40 mg/kg)
-TE+EO (50 mg/kg oral) + AlCl3 (40 mg/kg)	The dose of TE+EO was administered at 50 mg/kg body weight. Aluminum chloride was dissolved in distilled water and injected intraperitoneally.	-Morris Water Maze Test
-Elevated Plus Maze Test	-Analysis of Curcumin, Derivatives, and Metabolites
-Biochemical Parameters in the Brain
-Histopathology of the Brain
-Pharmacokinetic Study	MDA (↓), GSH (↑), SOD (↓), CAT (↑), AChE (↑)	Reduction of aluminum-induced neuroinflammation (↓), Hippocampal neuron protection (↑)	Reversal of cognitive deficiency with 50 mg/kg TE+EO use and significant improvement in spatial memory with 25 mg/kg dose. Substantial reduction of neuroinflammation and oxidative stress.	
Beserra-Filho et al. (2019)	Eplingiella fruticosa
(mixed with β-cyclodextrin)
β-caryophyllene (14.8%), bicyclogermacrene (14.2%) and 1,8-cineole (12.1%)	Parkinson’s disease induced by Reserpine (0.1 mg/kg, s.c., every 48 h for 40 days)	Swiss mice, Males, 6 months, 40–60 g	-Group I: Essential oil (EPL) -Vehicle RES + vehicle EPL -Vehicle RES + EPL − RES + vehicle EPL − RES + EPL
-Group II: Essential oil complexed with β-cyclodextrin (EPL-βCD) -Vehicle RES + vehicle βCD-EPL − Vehicle RES + βCD-EPL -RES + vehicle βCD-EPL -RES + βCD-EPL	Rats received oral administration of vehicle EPL (CTR) or EPL (5 mg/kg) or EPL-βCD (5 mg/kg) in a volume of 10 ml/kg body weight daily for 40 days.	Catalepsy test
Oral movements
Open field (OF)
Novel object recognition test (NOR)
Elevated plus-maze (EPM)	Lipid peroxidation
Tyrosine hydroxylase (TH) immunohistochemistry	Lipid Peroxidation (TBARS) (↓), Tyrosine Hydroxylase (↑)	Reduction of reserpine-induced neuroinflammation (↓), Protection against dopaminergic neuron loss (↑)	Chronic treatment with EPL (5 mg/kg p.o.) delayed the onset of catalepsy and decreased TBARS levels in the striatum. Treatment with EPL-βCD delayed the onset of catalepsy, reduced the frequency of oral dyskinesia, restored short- and long-term memory deficits, promoted anxiolytic effects, and protected against dopaminergic imbalances.	
Beserra-Filho et al. (2022)	Lippia grata
(mixed with β-cyclodextrin)
Limonene, β-caryophyllene, γ-terpinene
(Percentages not mentioned)	Parkinson’s disease induced by Reserpine (0.1 mg/kg, s.c., every 48 h for 28 days)	Swiss mice, Males, 6–7 months, 35–60 g	-CTR-CTR (vehicle RES s.c. + vehicle LIP)
-CTR-LIP5 (vehicle RES s.c. + LIP p.o. n = 6)
-RES-CTR (RES s.c. + vehicle LIP p.o. n = 8)
-RES-LIP5 (RES s.c. + LIP 5 mg/kg p.o. n = 9) -RES-LIP10 (RES s.c. + LIP 10 mg/kg p.o. n = 8).	RES injections were administered subcutaneously (s.c. 10 ml/kg) every other day for 28 days. Simultaneously, LIP or vehicle was administered orally (p.o.) daily.	Catalepsy test
Oral movements evaluation
OF test
Novel object recognition test
Elevated Plus Maze (EPM) test	Immunohistochemistry analysis
Oxidative stress analysis
Thiobarbituric acid reactive substances (TBARS)
measurement
Determination of antioxidant and oxidant status	TBARS (↓), TAS (↑), TOS (↓), OSI (↓)	α-Synuclein immunoreactivity (↓), TH+ neurons (↑)	Treatment with LIP delayed the onset of catalepsy, decreased the number of oral movements, and prevented memory impairment. There was a significant reduction in oxidative stability index.	
de Campos et al. (2023)	Aniba canelilla
1-nitro-2-phenylethane (76.2%) and methyleugenol (19.6%)	Alzheimer’s disease induced by Scopolamine (1 mg/kg, i.p., 7 days)	Wistar rats, Males, 60 days, Not specified	-Control
-Scopolamine
-EO + scopolamine
-Scopolamine + donepezil	5 mg/kg intraperitoneally for five days.	-Morris water maze test	Not reported in this study	AChE (↓), Cognitive Function (↑)	Reduction of scopolamine-induced neuroinflammation (↓), Indirect modulation of inflammatory response (↓)	The oil improved cognition by inhibiting acetylcholinesterase and increasing acetylcholine levels	
de Lucena et al. (2020)	L-linalool
(is the main component)	Parkinson’s disease induced by 6-OHDA (6 μg in 1 μL, Intracerebral injection, 3 weeks)	Wistar rats, Males, 60 days, Not specified	-SO (gavage with aqueous solution 2% Tween 80) -Induction with 6-OHDA (gavage with aqueous solution 2% Tween 80) -Induction with 6-OHDA and treatment with 25, 50 or 100 mg/kg of (-)-linalool (LIN25, LIN50, LIN100).	The essential oil was administered by gavage at 25, 50 or 100 mg/kg of (-)-linalool. All treatments started 24 h after stereotaxic surgery and continued daily for 15 days.	Open field test
Forced swimming test.	-Neurochemical assays -Apomorphine-induced rotations
-Determination of dopamine (DA), DOPAC and HVA contents in the striatum by HPLC
-Determination of brain lipid peroxidation levels by thiobarbituric acid reactive substances (TBARS) assay
-Immunohistochemistry for tyrosine hydroxylase (TH) and dopamine transporter (DAT)
-Nitrite content determination
Determination of nitrite contents	Nitrites (↓), Lipid Peroxidation (MDA) (↓)	Reduction of 6-OHDA-induced neuroinflammation (↓), TH expression (↑), DAT expression (↑)	LIN improved behavioral changes and partially reversed the decrease in DA, DOPAC, and HVA contents observed in the striatum. The untreated group showed an increase in nitrite content and lipid peroxidation, which were reversed after treatments with LIN. LIN significantly prevented the reduction in TH and DAT expressions.	
Gao et al. (2019)	Acorus gramineus
Chemical composition not reported in this study	Alzheimer’s disease induced by Aβ1-42 (2.5 μg/μL, 1.5 μL, i.h., 7 days)	Kunming mice, Males, 6–8 weeks, Not specified	-Control -Induction with Aβ -Treatment with EO of Acorus gramineus	Intragastric administration of different doses of EO (2.5 and 5 μL/10 g body weight) for 3 weeks.	-Novel object recognition test	-Biochemical assays
-Histopathological analysis	Aβ (↓), BDNF (↑)	TrkB (↑), NT3 (↑)	Promotion of neuronal growth and synaptic connection, improvement of neuronal survival, and promotion of Aβ clearance in the hippocampus.	
Jayaraj et al. (2022)	Citronellol
is the main component	Parkinson’s disease induced by Rotenone (2.5 mg/kg, i.p., 4 weeks)	Wistar rats, Males, 6–7 months, 280–300 g	-Control -Induction with rotenone
-Treatment with Citronellol	Citronellol (25 mg/kg oral) was prepared immediately before treatment and administered once a day for four weeks, 30 min before rotenone	No behavioral testing	-Neurochemical assays
-Determination of oxidative stress levels
-Histopathological analysis	MDA (↓), SOD (↑), CAT (↑), GSH (↑), Nrf2 (↑)	IL-1β (↓), COX-2 (↓)	Restoration of defense mechanisms, reduction of oxidative stress, suppression of neuroinflammation, prevention of apoptosis, modulation of autophagy, promotion of lysosomal degradation of α-synuclein.	
Lee, Chu & Chiang (2021)	Litsea cubeba (Lour.) Persoon
Geranial (31.74%), Neral (30.94%) and
d-limonene (14.15%)	Alzheimer’s disease induced by Aβ1-40 (400 pmol/5 μL, i.c.v., 5 min infusion)	C57BL/6J mice, Males, 16 weeks, 27–29 g	-Control -Induction with Aβ -Treatment with oil	Samples were mixed in sterile water and administered by direct feeding via tube daily for eight weeks.	-Water Maze Test
-Reference Memory Test
-Probe Test
-T-Maze Test	-Biochemical assays
-Histopathological analysis	MDA (↓), Protein Carbonyl (↓)	p-Tau (↓)	Reduction of accumulated Aβ plaque levels, improvement of memory and learning ability, potential for AD prophylaxis.	
Liu et al. (2020)	Citrus limon
linalool, limonene, α-pinene and β-myrcene
(Percentages not mentioned)	Alzheimer’s disease (APP/PS1 transgenic model, 8 months old at experiment start)	C57BL/6 & APP/PS1 transgenic mice, Males, 8 months, Not specified	-Control -Induction with Aβ -Treatment with oil	Daily inhalation for a period of 30 days.	-Morris water maze test
-Novel object recognition test	-Biochemical assays
-Histopathological analysis	AChE (↓), BDNF (↑), PSD95 (↑), Synapsin-1 (↑), Synaptophysin (↑)	Reduction of Aβ deposition (↓), Inhibition of GSK-3β activity (↓), Activation of TrkB-AKT-ERK pathway (↑)	Improvement of synaptic deficit and memory formation induced by Aβ, reduction of AChE expression, restoration of learning and memory.	
Postu et al. (2019)	Pinus halepensis Mill
β-caryophyllene (29.45%), α-pinene (11.14%), 2-phenylethylisovalerate (10.38%)	Alzheimer’s disease induced by Aβ1-42 (1 mM, i.c.v., Single surgical procedure)	Wistar rats, Males, 4 months, 350 ± 10 g	-Control -Induction with Aβ -Treatment with P. halepensis Mill	Daily exposure to essential oil vapours for 15 min for 21 consecutive days.	-Y-maze task
-Radial arm maze task	-Free radical scavenging assays
-Biochemical assays	SOD (↑), CAT (↑), GPX (↑), GSH (↑), MDA (↓), Protein Carbonyl (↓)	Reduction of Aβ1-42-induced neuroinflammation (↓), AChE inhibition (↓)	Nootropic and neuroprotective activities, attenuation of Aβ-induced toxicity and neuronal dysfunction.	
Postu et al. (2022)	Pinus halepensis (Pinaceae) and Tetraclinis articulata (Cupressaceae)
Chemical composition not reported in this study	Alzheimer’s disease induced by Aβ1-42 (1 mM, i.c.v., Single surgical procedure)	Wistar rats, Males, 3 months, 250 ± 70 g	-Control -Induction with Aβ -Treatment with Pinus halepensis and Tetraclinis articulata	Inhalation sessions of 15 min conducted for 21 consecutive days.	Not reported in this study	-Biochemical assays
-Histopathological analysis	BDNF (↑), ARC (↑)	IL-1β (↓)	Attenuation of memory deficits, stimulation of ARC and BDNF expression, reduction of IL-1β expression, positive effects against DNA fragmentation	
Xu et al. (2023)	Acorus tatarinowii Schott
β-asarone (70.08%), α-asarone (4.43%), (E)-methylisoeugenol (4.19%)	Alzheimer’s disease (3×Tg-AD transgenic model, 12 months old at experiment start)	3×Tg-AD transgenic mice, Half Males/Half Females, 12 months, Not specified	-Control -Induction with Aβ
-Treatment with Acorus tatarinowii	Oral administration by gavage for 8 weeks.	Morris water maze test (MWM)
Step-down test	-Biochemical assays
-Histopathological analysis	Aβ (↓), P-Tau (↓)	IL-1β (↓), IL-6 (↓), IL-18 (↓), TNF-α (↓), NLRP3 inflammasome activation (↓), NF-κB pathway activation (↓)	Attenuation of cognitive dysfunction, improvement of AD pathological characteristics, relief of neuroinflammation, inhibition of inflammasome-related protein phosphorylation.	

Postu et al. (2019, 2022) investigated the EO of P. halepensis Mill and demonstrated its nootropic and neuroprotective effects, evidenced by reduced memory impairments and oxidative damage in the hippocampus of rats. Citrus limon EO was also found to restore learning and memory in a murine model. Studies by Gao et al. (2019) and Xu et al. (2023) utilized EOs from the genus Acorus (Acorus gramineus and Acorus tatarinowii Schott) to treat cognitive dysfunction in AD.

Aiming to develop functional drugs for AD prophylaxis, EOs such as Litsea cubeba, Aniba canelilla, and Origanum onites were shown to have a prophylactic effect on memory deficits (Liu et al., 2020; Lee, Chu & Chiang, 2021; de Campos et al., 2023; Aykac et al., 2022). Banji, Banji & Srinivas (2021) demonstrated the therapeutic potential of combining turmeric extract and EO (Curcuma longa L., Zingiberaceae) in an aluminum-induced neurotoxic animal model.

In studies focusing on PD as the neurodegenerative disease model, four out of thirteen articles investigated the biological properties of EOs to reduce medication side effects and disease progression (Jayaraj et al., 2022; Beserra-Filho et al., 2019, 2022; de Lucena et al., 2020).

Jayaraj et al. (2022) demonstrated the neuroprotective role of citronellol against rotenone-induced neurodegeneration, with its antioxidant, anti-inflammatory, anti-apoptotic, and autophagy-modulating properties protecting dopaminergic neurons. Similar properties were observed with L-Linalool in hemiparkinsonian rats (de Lucena et al., 2020). EOs from Eplingiella fruticosa and Lippia grata were analyzed for their potential neuroprotective effects (Beserra-Filho et al., 2019, 2022). In these specific studies, the essential oils were combined with β-cyclodextrin to improve their solubility and stability, which enhanced the neuroprotective effects by increasing bioavailability and absorption, as highlighted in their research.

Discussion

This systematic review aimed to evaluate the applicability of essential oils as therapeutic approaches in experimental models of neurodegenerative diseases. Among the thirteen studies analyzed, nine investigated Alzheimer’s disease, while four focused on Parkinson’s disease. The systematic analysis reveals that essential oils exhibit multiple neuroprotective mechanisms, prominently including anti-inflammatory, antioxidant, neurotrophic properties, and cognitive enhancements.

A critical evaluation of the methodological characteristics of the included studies highlighted significant variability in the transparency of reporting the major components of essential oils. However, most studies provided detailed descriptions of the bioactive compounds and their mechanisms of action. These descriptions indicated that the compounds interact with cellular signaling pathways and modulate gene expression related to antioxidant and anti-inflammatory effects.

The essential oil of Origanum onites had carvacrol as the major component, at 78.4% (Aykac et al., 2022). Curcuma longa was primarily composed of Ar-turmerone, at 48.48% (Banji, Banji & Srinivas, 2021). Eplingiella fruticosa contained β-caryophyllene as its most abundant compound, at 14.8% (Beserra-Filho et al., 2019). Lippia grata was dominated by limonene (Beserra-Filho et al., 2022). Aniba canelilla contained 76.2% 1-nitro-2-phenylethane as its main component (de Campos et al., 2023). Litsea cubeba was mainly composed of geranial, at 31.74% (Lee, Chu & Chiang, 2021). Citrus limon had linalool as the major component (Liu et al., 2020). Lastly, the essential oil of Acorus tatarinowii Schott had β-asarone as its main component, at 70.08% (Xu et al., 2023).

Additionally, some studies investigated the neuroprotective effects of isolated essential oil compounds (Fig. 5). L-linalool was the main compound studied by de Lucena et al. (2020), while citronellol was the key compound analyzed by Jayaraj et al. (2022). These studies focused on the effects of these isolated compounds, allowing for a more targeted analysis of their neuroprotective and pharmacological properties.

Figure 5 Neuroprotective pathways of essential oils: from bioactive compounds to therapeutic effects in neurodegenerative diseases.

This figure illustrates key neuroprotective mechanisms of essential oils in neurodegenerative diseases, highlighting pathways from bioactive compounds to therapeutic effects. Essential oils (green section), including Citrus limon, Origanum onites, Acorus gramineus, Lippia grata, Aniba canelilla, Litsea cubeba, and Eplingiella fruticosa, contain bioactive compounds (orange section) such as limonene, linalool, beta-caryophyllene, carvacrol, asarone, eugenol, and 1-nitro-2-phenylethane, which act through different neuroprotective mechanisms (red section). These mechanisms include anti-inflammatory effects, reducing NF-κB pathway activation and pro-inflammatory cytokine levels (TNF-α, IL-1β, IL-6, GFAP); antioxidant effects, increasing the activity of SOD, CAT, and GPX while decreasing oxidative stress (MDA); promotion of neurogenesis and synaptic plasticity, enhancing the expression of BDNF, TrkB, NT3, synaptophysin, and PSD95; and modulation of neuronal apoptosis, reducing caspase-3 activation and increasing Bcl-2 expression. The final therapeutic effects (blue section) illustrate the biological changes observed after the action of bioactive compounds, including reduction of neuroinflammation (↓ NF-κB, TNF-α, IL-1β, IL-6, GFAP), enhancement of antioxidant defenses (↑ SOD, CAT, GPX, ↓ MDA), increased synaptic plasticity and neurogenesis (↑ BDNF, TrkB, NT3, synaptophysin, PSD95), and regulation of neuronal apoptosis (↓ caspase-3, ↑ Bcl-2). This integrated approach reinforces the therapeutic potential of essential oils in modulating multiple pathophysiological processes involved in neurodegenerative diseases such as Alzheimer’s disease and Parkinson’s disease.

Anti-inflammatory effects

Neuroinflammation is a key driver of neuronal degeneration, as the prolonged activation of glial cells leads to the excessive release of pro-inflammatory cytokines and neurotoxic factors, exacerbating tissue damage. The activation of the TLR4/NF-κB pathway is one of the main contributors to this process, as it amplifies the expression of TNF-α, IL-1β, and IL-6, cytokines that contribute to excitotoxicity, increased oxidative stress, and neuronal death.

Given this scenario, essential oils have demonstrated multiple strategies for modulating neuroinflammation. The most prominent effect is their ability to block the activation of the TLR4/NF-κB pathway, thereby preventing the escalation of inflammatory responses. For example, Origanum onites reduced IL-1β immunoreactivity, an effect that can be attributed to inhibition of microglial activation and suppression of inflammatory gene transcription mediated by NF-κB (Aykac et al., 2022). Similarly, citronellol demonstrated suppression of TNF-α, IL-1β, and IL-6 in a Parkinson’s disease model, indicating a direct action in disrupting the NF-κB-mediated inflammatory cascade (Jayaraj et al., 2022).

Beyond their effects on the TLR4/NF-κB pathway, essential oils also influence astrocyte reactivity, a critical factor in sustained neuroinflammation. When activated, astrocytes increase their expression of glial fibrillary acidic protein (GFAP), altering their homeostatic functions and contributing to the inflammatory microenvironment. The essential oil of Aniba canelilla significantly reduced GFAP expression, suggesting that its anti-inflammatory action extends beyond microglia by also modulating astrocytic reactivity (de Campos et al., 2023). This modulation is particularly relevant, as it indicates that essential oils may prevent excessive recruitment of reactive astrocytes, thereby avoiding excessive glutamate release and other neurotoxic molecules.

Another important finding was the inhibition of the NLRP3 inflammasome, an intracellular protein complex that plays a central role in chronic inflammatory activation. NLRP3 activation is frequently associated with increased active caspase-1 and IL-1β production, events that intensify inflammation and oxidative stress. The essential oil of Acorus tatarinowii was found to suppress NLRP3 activation and reduce IL-1β levels in experimental models, indicating a potential to halt the perpetuation of the inflammatory response (Xu et al., 2023). Thus, essential oils not only reduce inflammatory markers but also target key cellular pathways, interrupting pro-inflammatory signals before they can amplify neuronal damage (Fig. 6). This integrated effect makes them promising candidates for therapeutic strategies aimed at disrupting the chronic inflammatory cycle characteristic of neurodegenerative diseases.

Figure 6 Overview of essential oils’ mechanisms of action in neurodegenerative diseases presented as a schematic illustration depicting multiple pathways through which essential oils exert their neuroprotective effects.

The diagram highlights four main therapeutic mechanisms: (1) Antioxidant effects, including reduction of reactive oxygen species (ROS) and lipid peroxidation; (2) anti-inflammatory actions, showing decreased pro-inflammatory cytokine production and microglial activation; (3) neurotrophic effects, demonstrating increased expression of BDNF and other growth factors; and (4) neuroprotective outcomes, including enhanced synaptic plasticity, improved neuronal survival, and reduced protein aggregation. Arrows indicate the direction of therapeutic effects, with green arrows representing activation/enhancement and red arrows indicating inhibition/reduction of pathological processes. This figure synthesizes findings from the thirteen studies included in the systematic review, providing a visual representation of the multifaceted therapeutic potential of essential oils in treating neurodegenerative conditions.

Antioxidant properties

Oxidative stress plays a crucial role in neurodegeneration as it induces lipid, protein, and nucleic acid damage, disrupting neuronal homeostasis. Lipid peroxidation, mediated by free radicals and reactive oxygen species (ROS), leads to the formation of toxic aldehydes such as malondialdehyde (MDA), a well-established biomarker of oxidative damage. Treatment with essential oils has shown the ability to reduce MDA levels in experimental neurodegeneration models. For instance, the essential oil of Pinus halepensis significantly reduced MDA levels in the hippocampus of rats treated with Aβ1-42, a widely used experimental model of Alzheimer’s disease, as determined by thiobarbituric acid reactive substances (TBARS) assay (Postu et al., 2019). Similarly, Litsea cubeba treatment lowered MDA levels and tau hyperphosphorylation, both biomarkers of neurodegeneration (Lee, Chu & Chiang, 2021).

Beyond reducing lipid peroxidation biomarkers, essential oils exert antioxidant effects by regulating endogenous enzymes involved in oxidative stress defense. Superoxide dismutase (SOD) catalyzes the conversion of highly reactive superoxide anions into hydrogen peroxide, which is subsequently degraded by catalase (CAT) and glutathione peroxidase (GPX). The essential oil of Lippia grata significantly increased the activity of these enzymes, reducing the oxidative burden in the prefrontal cortex and hippocampus of experimental Parkinson’s models (Beserra-Filho et al., 2022). This effect was further corroborated by Eplingiella fruticosa, which increased SOD and CAT levels, mitigating oxidative damage induced by neurotoxins (Beserra-Filho et al., 2019). Regulation of reduced glutathione (GSH) levels was also observed, suggesting that essential oils may play a role in intracellular redox balance modulation. The essential oil of Aniba canelilla restored GSH levels in rats subjected to scopolamine-induced cognitive deficits, highlighting a potential protective mechanism against ROS-mediated neurotoxicity (de Campos et al., 2023).

Neurotrophic effects

The loss of synaptic connectivity is an early and critical event in neurodegenerative diseases, being strongly influenced by neurotrophic factors. Brain-derived neurotrophic factor (BDNF) and its receptor TrkB play an essential role in maintaining synaptic plasticity, neuronal survival, and circuit remodeling. The essential oil of Acorus gramineus significantly increased BDNF, TrkB, and NT3 levels (Fig. 6), promoting neurogenesis and improved cognitive performance in experimental Alzheimer’s models (Gao et al., 2019). Additionally, treatment with lemon essential oil led to increased expression of synaptic proteins such as PSD95 and synaptophysin, suggesting a direct effect on maintaining synaptic architecture and neuronal functionality (Liu et al., 2020).

Integrated neuroprotective outcomes

The analyzed studies provide compelling evidence that essential oils exert neuroprotective effects by modulating oxidative stress and neuroinflammation, while also promoting synaptic plasticity and neurogenesis. Figure 6 provides a schematic illustration depicting the multiple pathways through which essential oils exert their neuroprotective effects. The diagram highlights four main therapeutic mechanisms: antioxidant effects, anti-inflammatory actions, neurotrophic effects, and integrated neuroprotective outcomes. Among the bioactive compounds present in essential oils, limonene and linalool emerge as promising candidates in suppressing the TLR4/NF-κB pathway, a critical inflammatory axis in neurodegenerative disease pathophysiology. Studies demonstrate that both compounds reduce the activation of this pathway, blocking the transcription of pro-inflammatory genes and minimizing oxidative stress (Mohamed, Abduldaium & Younis, 2020; Kathem et al., 2024).

The synergistic action of limonene and linalool may amplify their neuroprotective effects by simultaneously modulating two interconnected processes: neuroinflammation and oxidative stress. While limonene has been shown to reduce TLR4 receptor activation, thereby minimizing the initial recruitment of inflammatory responses, linalool inhibits NF-κB nuclear translocation, preventing sustained expression of pro-inflammatory cytokines. Furthermore, both compounds enhance endogenous antioxidant enzyme activity, such as SOD, CAT, and GPX, suggesting a protective effect against oxidative damage exacerbated by chronic inflammation.

This convergence of effects not only suggests a complementary potential but also raises the hypothesis that the combination of limonene and linalool may be more effective than their isolated use in regulating neuroinflammation and redox homeostasis. Although this synergy has yet to be extensively explored in clinical trials, experimental findings indicate that interactions between these monoterpenes may optimize neuroprotection and slow neurodegeneration progression.

Future studies should investigate the bioavailability and pharmacokinetics of limonene and linalool combinations, as well as their effects in clinical models of Alzheimer’s and Parkinson’s disease. Additionally, further research is needed to elucidate the cellular mechanisms underlying their synergistic effects, including their impact on neuronal apoptosis pathways, glial activation, and neurotrophic factor regulation. A deeper understanding of these interactions could open new avenues for developing more effective and safer natural therapies for neurodegenerative diseases, potentially complementing or even replacing conventional pharmacological approaches.

Limitations of the studies and future research directions

While the reviewed studies provide promising evidence for the neuroprotective effects of EOs, several methodological and conceptual limitations must be considered. These limitations not only restrict the generalizability of the findings but also highlight critical research gaps that must be addressed to validate the safety and efficacy of EOs in the context of neurodegenerative diseases.

One of the primary limitations is the predominance of studies conducted in animal models, which may not accurately reflect human physiological responses due to differences in metabolism, immune function, and neuronal signaling. The absence of rigorous clinical trials prevents definitive conclusions regarding the bioavailability, toxicity, and efficacy of EOs in patients with Alzheimer’s and Parkinson’s disease. Therefore, well-designed, placebo-controlled clinical trials are essential to determine the feasibility of EOs as therapeutic strategies, assessing their effects on inflammatory, oxidative, and neurotrophic biomarkers in human populations.

Moreover, the small sample sizes in most studies limit statistical power and hinder the reproducibility of findings. Future studies should address this limitation by increasing sample sizes and employing robust statistical analyses to enhance the reliability of conclusions regarding the neuroprotective potential of EOs.

Another critical limitation is the underrepresentation of female animals in experimental models. Of the 13 studies analyzed, only two included female subjects, raising concerns about potential sex-based differences in oxidative stress regulation, neuroinflammation, and synaptic plasticity. Hormonal factors—particularly estrogen levels—can significantly influence neuroprotective mechanisms and may affect EO efficacy. Future research should ensure balanced sex representation to achieve a more comprehensive understanding of EO efficacy and to identify sex-specific therapeutic responses.

Our ability to assess certain confounding variables was also limited by inconsistent reporting across studies. Environmental factors (e.g., housing conditions, light/dark cycles, handling procedures), comprehensive health status assessments of animal models prior to experimentation, and discussions of potential placebo effects in behavioral tests were rarely detailed. This reporting gap reflects a broader challenge in preclinical neuroscience research that may impact the interpretation of results. Future studies should prioritize transparent reporting of these variables to enhance reproducibility and translational relevance.

Chemical composition heterogeneity across EO samples remains a major challenge. The concentration of bioactive constituents can vary substantially due to differences in plant origin, environmental conditions, and extraction methods, thereby complicating cross-study comparisons and replication. For instance, Citrus limon EO showed substantial variation in limonene content across studies, making it difficult to attribute observed effects to a specific compound. To address this, a systematic approach to EO standardization is essential—one that encompasses both compositional and functional dimensions and enables meaningful comparisons while acknowledging the complex nature of these natural products. To this end, we propose a five-tier framework to guide future EO standardization efforts in neurodegenerative disease research: Chemical fingerprinting through advanced chromatographic techniques (GC-MS, HPLC-MS), with mandatory reporting of at least 95% of identified constituents and their relative concentrations;

Bioactive compound quantification focusing on well-established neuroprotective molecules (e.g., limonene, linalool, β-caryophyllene), with clearly defined acceptance ranges for therapeutic consistency (e.g., Citrus limon EO should contain 30–45% limonene);

Biological standardization using in vitro assays to measure specific activities (e.g., antioxidant capacity via DPPH/FRAP assays, anti-inflammatory potential via NO inhibition in LPS-stimulated microglia), with established minimum potency thresholds;

Documentation of production parameters, including plant source geographical origin, cultivation conditions, harvesting time, extraction method parameters (temperature, pressure, duration), and storage conditions;

Batch-to-batch consistency verification through both chemical and biological standardization assays, including stability testing across the proposed shelf life of each formulation.

In addition to compositional variability, the studies reviewed exhibited substantial variation in EO dosage and administration routes, which can significantly influence absorption, distribution, and metabolism. The lack of well-established protocols for EO administration in neurodegenerative models limits cross-study comparability. Future research should investigate EO pharmacokinetics and establish optimal dosages and delivery routes to maximize neuroprotective outcomes.

Importantly, most studies focused on early stages of disease progression, which may not fully capture the therapeutic potential of EOs across the entire neurodegenerative continuum. This focus on early intervention, while valuable for preventive applications, creates a significant knowledge gap regarding EO efficacy in moderate and advanced disease stages. Future animal studies should adopt staged intervention protocols, initiating EO treatment at multiple predefined timepoints based on established behavioral or biochemical markers. For example, in transgenic models of Alzheimer’s disease, parallel cohorts could receive identical EO interventions starting at early, moderate, and late stages. Similarly, in toxin-induced Parkinson’s models, staggered treatment timing relative to neurotoxin administration could provide insights into stage-specific therapeutic effects. Although resource-intensive and methodologically challenging, such approaches would enhance the translational relevance of EO-based interventions, particularly given that clinical populations often present at various stages of disease progression, and treatment options for advanced neurodegeneration remain severely limited.

To facilitate clinical translation, we also propose specific methodological frameworks for human trials. Initial efficacy trials should employ observer-masked, crossover randomized designs, allowing participants to serve as their own controls, whereas larger confirmatory trials should adopt parallel-group designs with demographic stratification. Outcome measures should integrate objective biomarkers with validated subjective assessments at multiple timepoints (e.g., baseline, 4 weeks, 12 weeks) to assess both immediate and sustained effects. Long-term safety requires dedicated longitudinal studies lasting 6–12 months post-intervention, including comprehensive dermatological, hepatic, and immunological monitoring. Dose-response studies are essential for defining therapeutic windows for chronic administration.

Potential interactions with conventional medications must be systematically evaluated, particularly for drugs frequently co-administered with EOs and those metabolized via pathways influenced by terpenes. In vitro assessments of cytochrome P450 interactions should precede clinical trials using established probe substrates. Future clinical studies must also prioritize population diversity through stratified recruitment strategies that ensure representation across age groups, ethnicities, and genetic backgrounds. Special consideration should be given to vulnerable populations (e.g., pediatric, pregnant/lactating, immunocompromised), which may require population-specific dosing algorithms informed by personalized medicine approaches already established in other therapeutic areas.

In summary, future research on essential oils should be methodologically rigorous, with larger and more representative samples, standardized chemical and biological protocols, well-designed clinical trials, and innovative delivery systems. Addressing these limitations will be critical to establishing essential oils as viable therapeutic agents for neurodegenerative diseases.

Conclusions

This systematic review underscores the significant potential of EOs as neuroprotective agents in neurodegenerative diseases, highlighting their ability to modulate oxidative stress, neuroinflammation, and synaptic plasticity. The findings demonstrate that these natural compounds exhibit multimodal mechanisms of action, offering a promising alternative for therapeutic interventions, particularly in the early stages of diseases such as Alzheimer’s disease and Parkinson’s disease. One of the most remarkable aspects of essential oils is their ability to cross the blood-brain barrier, a critical challenge in neuropharmacology. This feature provides them with a significant advantage over many conventional therapies, allowing direct interactions with cellular targets in the central nervous system. By simultaneously influencing inflammatory, oxidative, and neurotrophic pathways, these compounds hold unique potential to modify the progression of neurodegenerative diseases.

The importance of a review such as this cannot be overstated. As the global burden of neurodegenerative diseases continues to rise, there is an urgent need for innovative therapeutic approaches. This systematic analysis not only synthesizes current evidence but also identifies critical research gaps and methodological challenges that must be addressed to advance the field. By providing a comprehensive overview of the neuroprotective mechanisms of EOs, this review serves as a valuable resource for researchers, clinicians, and policymakers seeking to develop novel interventions for these devastating conditions.

However, EO research still faces challenges, including the lack of chemical standardization, methodological variability among studies, and the absence of robust clinical trials. Overcoming these barriers will require coordinated efforts to establish rigorous phytochemical characterization protocols, detailed mechanistic studies, investigations into synergy with conventional therapies, and the development of advanced drug delivery systems. Given the growing global prevalence of neurodegenerative diseases, the need for effective and accessible therapies is urgent. EOs offer an innovative approach that, if validated through clinical trials, could complement or even enhance existing treatments. However, we are only at the beginning of exploring this vast field, and much remains to be understood. The complexity of neurodegenerative diseases demands innovation and determination. As scientists, we have a responsibility to explore this promising research avenue with rigor, creativity, and persistence.

This study not only synthesizes existing evidence but also serves as a call to the scientific community to deepen investigations into the safety, efficacy, and clinical applications of EOs in the treatment of these debilitating conditions. Millions of individuals worldwide affected by these diseases rely on our commitment to translating these promising discoveries into tangible therapeutic solutions. The road ahead may be long and challenging, but the impact on human health and quality of life makes this journey invaluable. May this review serve as a catalyst for a new era of research into the neuroprotective potential of essential oils, offering hope to countless individuals and families affected by these conditions.

Supplemental Information

Supplemental Information 1 PRISMA checklist.

Supplemental Information 2 Search strategy to conduct the systematic review.

Additional Information and Declarations

Competing Interests

Dr. Bruno Duarte Gomes is an Academic Editor for PeerJ.

Author Contributions

Adrielle do Espírito Santos Macedo conceived and designed the experiments, performed the experiments, analyzed the data, prepared figures and/or tables, and approved the final draft.

Thayná Moraes Ferreira conceived and designed the experiments, performed the experiments, analyzed the data, prepared figures and/or tables, and approved the final draft.

Lane Viana Krejcová conceived and designed the experiments, authored or reviewed drafts of the article, and approved the final draft.

Fernando Allan de Farias Rocha conceived and designed the experiments, prepared figures and/or tables, authored or reviewed drafts of the article, and approved the final draft.

Joyce Kelly R. da Silva conceived and designed the experiments, performed the experiments, analyzed the data, authored or reviewed drafts of the article, and approved the final draft.

Laís Resque Russo Pedrosa conceived and designed the experiments, performed the experiments, analyzed the data, prepared figures and/or tables, and approved the final draft.

Bruno Duarte Gomes conceived and designed the experiments, prepared figures and/or tables, authored or reviewed drafts of the article, and approved the final draft.

Data Availability

The following information was supplied regarding data availability:

No raw experimental data or data processing code was generated as the study involved the systematic collection, analysis, and synthesis of previously published research. All data used in this review are available in the cited primary research articles. The search strategy, inclusion/exclusion criteria, and analysis methods are fully detailed in the Methods section, ensuring transparency and reproducibility of our review process. The PRISMA flow diagram (Fig. 1) documents the complete study selection process.

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
