# Peer review of "Neuroprotective effects of essential oils in animal models of Alzheimer’s and Parkinson’s disease: a systematic review"

_PeerJ, doi:10.7717/peerj.19643_

## Round 0.1 · original submission · Major Revisions

Dear Co-Authors,

We have received a “Major Revisions” evaluation for your manuscript, with detailed feedback highlighting its strengths and areas for improvement. To address the reviewers’ concerns, you need to restructure the results table to include animal data, PD models, and key outcomes for clarity, expand the discussion by categorizing findings based on neuroprotective mechanisms and incorporating visual representations such as pathway diagrams, and add a dedicated section on study limitations and specific future directions. Reviewer 2 emphasizes the need to address the variability in essential oil compositions and propose a framework for standardization while also elaborating on the synergy between key compounds (e.g., α-pinene, limonene) and their neuroprotective mechanisms. You must improve figure quality (Figures 2 and 5), enhance grammatical precision, and refine the manuscript’s structure for better readability. By thoroughly addressing these points and responding to all reviewer comments, you can significantly strengthen the manuscript and improve its likelihood of acceptance.

Kind regards,
Dr. Manuel Jiménez

Reviewer 1 ·

Basic reporting

no comment

Experimental design

-elaborate on the research question and the current literature gap.
-The paper's methodology aligns well with systematic review standards and is executed well.

Validity of the findings

Result -Table:
The result table is somewhat challenging to follow. I suggest restructuring it to include details such as animal data (species, sex, n, weight, age), PD model (toxin, dose, duration, route of administration), and key outcomes (behavioral tests, dopaminergic damage, anti-inflammatory and antioxidant properties, etc) which could make the results more comprehensive and precise. This format would facilitate easier understanding and comparison.

-Consider mentioning specific markers for oxidative stress / inflammatory markers (Eg TNFa, SOD, etc) to provide more detailed insights.

Discussion:
The discussion could be more elaborative and structured to aid reader comprehension. Categorizing the findings based on mechanisms underlying the neuroprotective effects would make it clearer. rather than just state the finding, elaborate and compare. Additionally, including a visual representation of these mechanisms, such as a pathway diagram, would significantly improve the reader's understanding.

Study Limitations:
It would be beneficial to include a dedicated section on study limitations to address gaps and provide context for the findings.

Future Directions:
The suggestions for study limitation & future research are currently too general. I recommend being more specific, tailoring them to the findings and gaps identified in this review.

Figures and Visuals:
The image in Figure 2 is adequate in terms of content but needs improvement in quality to ensure clarity
Fig 5 is overly simplfied-include specific pathway/mechanism to give the fig more value

Additional comments

you can refer to this paper to improve the discussion section - https://doi.org/10.1007/s12640-021-00450-x)

·

Basic reporting

This systematic review on the neuroprotective properties of essential oils (EOs) in neurodegenerative diseases presents a comprehensive overview of recent advancements in this area.
While the review highlights thirteen high-quality in vivo studies, this number is relatively small. A broader analysis could yield more robust conclusions.
References OK

Experimental design

The predominant use of animal models (primarily Wistar rats and various mouse strains) raises concerns regarding the translatability of results to humans. Differences in metabolism, physiology, and complexity of human neurological disorders may limit the applicability of animal study findings.
The review notes a wide range of dosing (5 to 100 mg/kg) and various administration routes, complicating comparisons across studies. This variability can lead to inconsistent results, making it difficult to establish optimal dosing guidelines for potential clinical applications.
The absence of standardized EO compositions presents a significant challenge. Different extraction methods, plant sources, and chemical profiles can lead to variations in efficacy. The review suggests that standardization is necessary but does not provide a clear framework for how this can be achieved.
Although the review identifies specific compounds (e.g., α-pinene, limonene) responsible for therapeutic effects, it lacks a thorough discussion on how these compounds function synergistically within the complex mixtures of EOs. A more nuanced understanding of the interplay between compounds could provide deeper insights into their neuroprotective mechanisms.
The paper does not adequately address potential confounding factors in the studies analyzed, such as the impact of environmental conditions, the health status of the animal models, and the potential for placebo effects in behavioral assessments.
While the manuscript calls for human clinical trials, it lacks specificity regarding the design of such studies. Future research should also address the long-term effects and safety of EOs, potential interactions with conventional medications, and the implications of using EOs in diverse populations.
The focus on early-stage intervention may overlook the potential benefits of EOs in later stages of neurodegenerative diseases. A more balanced approach examining the efficacy of EOs throughout the disease continuum could provide a more comprehensive understanding.

Validity of the findings

Considering that this is a systematic review, the term novelty is not applied.

---

## Round 0.2 · Minor Revisions

Dear Co-Authors:

Please consider the reviewer comments.

Thank you and best regards.

Dr. Manuel Jiménez

Reviewer 1 ·

Basic reporting

Discussion: This section shows noticeable improvement from the initial version. To strengthen the discussion further, consider comparing the EO’s mechanism of action with existing treatments for Parkinson’s disease (PD) or Alzheimer’s disease (AD). Does the EO mimic the current mechanisms of action (MOA), or does it act on alternative pathways? Highlighting this could provide insight into whether the EO is better positioned as a complementary or alternative therapy to existing treatment options.

Experimental design

no comment

Validity of the findings

Conclusion: Some portions of the conclusion contain filler words that dilute the impact of the message. Aim for a more concise and focused summary that emphasizes the key findings and their potential implications.

Additional comments

Formatting: All specific names of essential oils (EOs) should be italicized consistently throughout the manuscript for clarity and adherence to scientific writing conventions.

Figure : The figure appears overly simplified. In the main text, readers are directed to refer to Figure, but there is little substantive information to observe. Consider revising or omitting the figure if it does not significantly contribute to data interpretation or enhance understanding.

---

## Round 0.3 · Minor Revisions

As you will see, the reviewer has one minor comment about the scientific formatting of plant names. Please fix this and resubmit.

Reviewer 1 ·

Basic reporting

no comments

Experimental design

no comments

Validity of the findings

no comments

Additional comments

Thank you for the opportunity to review the revised manuscript. I am pleased to note that all the comments have been thoroughly addressed. The addition of a comparative discussion on the mechanism of action of essential oils versus conventional therapies has notably strengthened the manuscript, adding depth and clarity to the discussion section.

One minor point: please ensure that all plant names and essential oil names are italicized according to standard scientific formatting conventions

I recommend the manuscript for acceptance after this minor correction.

---

## Round 0.4 · accepted · Accept

The authors have addressed the reviewer’s comment and the article is now ready for publication.